# Prostate Cancer-Associated miRNAs in Saliva: First Steps to an Easily Accessible and Reliable Screening Tool

**DOI:** 10.3390/biom12101366

**Published:** 2022-09-24

**Authors:** Christoph Luedemann, Jan-Ludwig Reinersmann, Claudia Klinger, Stephan Degener, Nici Markus Dreger, Stephan Roth, Michael Kaufmann, Andreas Savelsbergh

**Affiliations:** 1Department of Internal Medicine, Gastroenterology, Hepatology, Infectiology, Nephrology, Endocrinology & Diabetology, University Hospital of Bonn, 53127 Bonn, Germany; 2Division of Functional Genomics, Chair for Biochemistry and Molecular Medicine, University of Witten/Herdecke, 58453 Witten, Germany; 3Department of General, Visceral and Vascular Surgery, Marienhospital Brühl, 50321 Brühl, Germany; 4Center for Biochemical Education and Research (ZBAF), University of Witten/Herdecke, 58453 Witten, Germany; 5Department of Urology, Helios University Hospital Wuppertal, University of Witten/Herdecke, 42283 Wuppertal, Germany

**Keywords:** salivary exosomes, miRNA, biomarkers, prostate cancer, hsa-mir-331-3p, hsa-mir-200b, circulating miRNA

## Abstract

Background: Common diagnostic tools for prostate cancer—prostate-specific antigen and transrectal biopsy—show only low predictive value and poor sensitivity. This study examines circulating miRNA in saliva to explore the possibility of a non-invasive and easy-to-execute diagnostic tool for prostate cancer screenings. Methods: 16 miRNAs were extracted from salivary exosomes and analyzed via the delta-CT method. The presented method enables an application of the test in any health institution and even outpatient sector. Recruited participants were suspected to suffer from prostate cancer due to elevated PSA serum levels. Of these participants, 43 were diagnosed with prostate cancer, while 31 suffered from benign diseases and served as control group. Results: hsa-mir-331-3p and hsa-mir-200b were significantly reduced in prostate cancer patients compared to the control group. ROC curve analysis revealed a reliable differentiation strength (AUC > 0.6) for both miRNAs with positive predictive values of 71% indicating prostate cancer. Differentiation of both groups based on PSA serum measurements was insufficient. The other 14 examined miRNAs showed no significant group differences. Conclusions: The presented method and miRNA are promising non-invasive tools to augment the current prostate cancer screening, thereby improving screening sensitivity and reducing numbers of false positive cancer suspects admitted to further invasive diagnostic and therapeutic steps.

## 1. Introduction

### 1.1. Background

Today, prostate cancer is one of the most common cancer types among men [1]. Worldwide, an estimated 1.1 million men were diagnosed with prostate cancer in 2020 and prostate cancer incidence varies more than 25-fold worldwide; the rates are highest in Australia/New Zealand, Northern America and in Western and Northern Europe. Lung cancer was the most frequent cause of death from cancer in 13 regions of the world, followed by prostate cancer [1]. Early detection is key in successfully treating prostate cancer. Preventive cancer screening can be an appropriate concept if the tools are right [2]. Diagnostic tests for prostate cancer usually involve blood level testing of prostate-specific antigen (PSA) and transrectal, ultrasound-guided biopsy of the prostatic gland. However, both diagnostic tools often fail to satisfy due to low predictive value, poor sensitivity and a high level of invasiveness but remain without any serious alternative [3,4,5,6]. PSA is not specific for this malignancy since conditions such as bacterial prostatitis, acute urinary retention, benign prostatic hyperplasia (BPH), trauma and physical manipulation can also falsely elevate serum PSA levels [7]. Elevated PSA levels do not correlate closely with disease severity; approximately 30% of people with PSA 5–10 and >50% with PSA > 10 will have prostate cancer. Conversely, about 10–15% of people with PSA < 5 will actually have prostate cancer [8]. The difficulty of diagnosing prostate cancer originates from its diverse appearance. The disease varies individually from latent and slow growing to aggressive and rapidly progressing lethal tumors with only little success of therapy when detected too late. A possibility to increase prognostic chances is to improve early detection.

Circulating miRNAs offer hope to overcome many drawbacks by virtue of their cancer-specific expression and easy accessibility in a variety of human body fluids such as blood/serum, urine, and saliva [9,10,11,12]. Over the last decades, miRNAs have emerged as potent biomarkers that are associated with gene expression in humans, making them available for diagnostic processing [13]. Non-coding RNA molecules are classified into three categories depending on their origin, molecular structure, associated proteins, and biological functions: short interfering RNAs (siRNAs), microRNAs (miRNAs) and piwi-interchanging RNAs (piRNAs). siRNA and miRNA are phylogenetically the most common in the human genome. miRNA and siRNA act in both somatic and germline lineages in a broad range of eukaryotic species to regulate endogenous genes and to defend the genome from invasive oligonucleotides [14]. MicroRNAs (miRNAs) are a class of small (approx. 22–25 nucleotides) regulatory, non-coding RNAs [15]. miRNAs appear in two different forms. One form is the extracellular stable form, which made them initially and especially interesting as biomarkers for cancer and other diseases. Some miRNAs are not extracellular and freely circulating, but loaded into exosomes or microvesicles [16]. Lawrie et al. (2008) were the first to show the presence of miRNAs in body fluids of patients suffering from B-cell lymphoma [17]. Other studies demonstrated that tumor-derived miRNAs enter the circulation even when originated from epithelial cancer types [18].

In addition, studies showed different levels of circulating miRNAs between tumor patients and healthy control groups. They identified specific expression patterns of serum miRNAs for lung cancer, colorectal cancer, and diabetes, providing evidence that serum miRNAs contain fingerprints for various diseases [19]. Several studies investigated miRNA alteration as a potential diagnostic and prognostic tool. The first attempt to delineate a prostate cancer-specific miRNA expression signature was published in 2006 [20]. Much effort has been invested in the search for possibilities of substituting the work on prostate tissue samples towards an elegant and non-invasive procedure such as the analysis of a patient’s serum or plasma. miRNAs play an important role in the substitution of, e.g., digital rectal examination (DRE) and PSA screening. The initial study was able to show a correlation between miRNAs found in plasma and the presence of prostate cancer [18].

A signature including miRNA-375 was validated by more than one group, making it a reliable marker of a systemic disease. Additionally, miRNA-375 reached a level of significance when correlated with tumor stage and Gleason score in patients’ sera [21].

Six upregulated miRNAs were identified in urine, making it possible to discriminate between PCa and BPH patients. miRNA-484 is among these identified molecules [22].

Despite the effort, to be able to discriminate men with potential PCA risk with a quick, easy and noninvasive test, this field needs further development [23]. Saliva screening still lacks a consensus signature that could be applied to the routine screening setting. Recent studies in this field of research lead to the selection of nine promising miRNA candidates for a salivary detection of prostate cancer [23].

It was found that therapy-induced neuroendocrine prostate cancer is increasing in incidence with the widespread use of highly potent androgen receptor-pathway inhibitors, most probably through neuroendocrine differentiation. This differentiation seems to be accompanied by key miRNA alterations including downregulation of miR-106a~363 cluster and upregulation of miR-301a and miR-375, as Bhagirath et al. demonstrated most recently [24]. New technologies have unveiled large numbers of valuable salivary biomarkers for different systemic diseases including cancer, based on the fact that saliva contains most of the serum constituents [25,26,27].

This study aimed to explore the diagnostic and clinical potential of miRNA in saliva samples as a new and easy to access screening tool to identify men suffering from prostate cancer. Due to the diversity of the molecules under consideration, a thorough literature review was performed to identify the most promising miRNA candidates.

### 1.2. Literature Review

We identified 16 circulating miRNAs with promising diagnostic relevance as possible biomarkers from whole saliva samples. Literature research was done according to the Preferred Reporting Items for Systematic Review and Meta-analysis statement [28]. We focused on publications with results obtained from body fluid examination and measurements and a linkage to prostate cancer. Four central publications reported either up- or downregulation of different miRNAs in body fluids (e.g., urine, blood, plasma) of prostate cancer patients: Bryant et al. (2012), Brase et al. (2011), Chen et al. (2012) and Moltzahn et al. (2011) [19,29,30,31]. Examination of these sequences in saliva samples has not been done before. Table 1 presents each source of literature and the corresponding miRNAs withdrawn from these sources. For the development and adaptation of the methodological concept of isolating the exosomes and extracting the miRNA, the publication of Gallo et al. [16] was of central importance.

### 1.3. Specimen Collection and Methodological Adaptions

The concept of extraction of miRNA from salivary exosomes was based on the model of Gallo et al. [16], see Figure 1. The protocol of saliva collection, storage and RNA isolation is designed to ensure the examination of only extracellular circulating exosomes secreted through the saliva glands into the saliva fluid. The sampling process and direct processing of the specimens is very difficult to implement in everyday clinical life or outpatient sectors. Above all, an immediate ultracentrifugation or execution of meticulous preparation steps demand specific equipment and training which are not available to every institution, practice or at home. With the modification of freeze storing the saliva samples right after saliva sampling as described and published by Wiegand et al. [32], specimens are much easier to handle. In this way, saliva samples can be collected and frozen in any health institution or at home and further transported and processed in centralized laboratories. This comes at the cost of freeze thawing oral tissue cells, food debris and other organic residues and with it releasing other exosomes/microvesicles into the solution. However, focusing the miRNA analysis on the above-mentioned prostate cancer associated sequences should provide reliable diagnostic specificity and eliminate the expected contamination and interferences to a minimum. A detailed description of the procedure follows in the next section.

## 2. Materials & Methods

### 2.1. Research Subjects

Patients were continually recruited from the urology department of the Helios University Hospital, Wuppertal. A signed informed consent for the study was mandatory. The conduction of the study took place from summer 2014 until winter 2018. Included were men of any age with elevated PSA-blood serum levels suspected to suffer from prostate cancer and admitted for result clarification via prostate biopsy. Exclusion criteria were secondary diagnosis, e.g., infectious diseases (HIV, hepatitis, tuberculosis etc.), Sicca Syndrome, prior interventions such as operations of the salivary gland or prostate, urinary diversion with intestine tissue, radiotherapy of the pelvic region or antiandrogen therapy. The sample size of the cohort was planned to be 100 participants with equal distribution between diagnosed prostate-cancer patients and non-cancer patients. Men with elevated PSA levels without histologic proof of cancer in transrectal prostate biopsy samples (31 participants) were assigned to the no-cancer control group. Besides prostate cancer, other reasons for PSA level elevation were medical issues such as prostatitis, urinary occlusion, urinary tract infection or secondary genesis, but most commonly benign prostatic syndrome [33].

In all cases routine blood and urine samples were taken, a routine urine bacterial culture was acquired and before prostate biopsy preventive antibiotic therapy (Ciprofloxacin) was given. Each patient gave an additional 5 mL whole saliva sample. Patients were instructed to refrain for two hours from smoking, drinking or eating to keep all samples pure, genuine and undiluted. In total, the study involved 86 patients of which 74 patients were able to provide whole saliva samples valid for miRNA concentration level analysis. A total of 43 patients were later diagnosed with histologically verified prostate cancer. Furthermore, 31 patients missed the histological criteria for prostate cancer, presenting other causes for elevated PSA levels, e.g., benign prostatic syndrome. The diagnoses behind these causes were not itemized any further. This cohort represents the study sample for analysis of concentration differences of 16 miRNAs between cancer and non-cancer patients. Only the two authors, both medical doctors responsible for specimen collection, had initial access to personal information of the patients. The collected data were anonymized and retrospective conclusions to single patients were and are not possible.

Immediately after collection, the saliva samples were frozen at −20 °C in a conventional freezer and later stored at −80 °C in our laboratory facilities. For analysis, the samples were thawed at room temperature and centrifuged at 1000× *g* for 2 min to remove food and cell debris. Two milliliters of the supernatant were centrifuged at 1500 rpm for 10 min (Mini-Spin); additionally, for saliva with high viscosity, it was followed by a centrifugation at 4700 rpm for 10 min (Mini-Spin). The supernatant was then transferred into a new tube and centrifuged at 13,500 rpm for 15 min (Mini-Spin) at first and then at 160,000× *g* for 60 min at 4 °C in a SS55S Rotor in an MGX-120 Ultracentrifuge (Sorvall). The supernatant was then discarded. One mL of Trizol reagent was added to the pellet and mixed. Then, 200 µL of chloroform were added to the Trizol and vortexed for 30 s, before the mixture was allowed to rest for 3 min and subsequently centrifuged at 13,500 rpm for 15 min (Mini-Spin). The upper, clear phase was transferred into a new Eppendorf tube and 500 µL of Isopropanol was added. The mixture was then incubated at −20 °C for 20 min. After that, the mixture was centrifuged at 13,500 rpm for 15 min (Mini-Spin) and the supernatant was again discarded. The pellet was washed with 1 mL of 75% ethanol, followed by centrifugation at 11,000 rpm for 5 min (Mini-Spin) and discarding of the supernatant. The pellet containing the miRNA was dried for 20 min with the cover lid opened at room temperature, and 25 µL of H_2_O (RNAse-free) were added to the pellet followed by vortexing. At last, the mixture was incubated at 60 °C for 15 min, followed by a brief centrifugation due to condensed water at the cover lid.

### 2.2. qRT-PCR

First, miRNA was transcribed into cDNA via reverse transcriptase (RT) reaction using the miScript II RT Kit 50 from Qiagen^®^ was used. Quantitative real-time PCR (qRT-PCR) was used to detect and quantify sequences of interest. In total, 8 µL Master Mix were combined with 12 µL template RNA, adding up to 20 µL RT-mix for the reverse transcriptase reaction protocol.

For the qRT-PCR, the *miScript SYBR Green PCR Kit* from Qiagen^®^ and specific primer assays for the sequences were used. The standardizing assay (internal control gene) was U6 snRNA, which by the time this study was conducted was seen to be one of the state-of-the-art reference genes for this type of investigation and its reliability was proven by former studies [34,35,36]. qPCR cycling conditions were used as described in the *miScript SYBR Green PCR Kit* handbook for the detection of mature miRNA. The data from qRT-PCR was documented and processed with qPCRSoft 21 program. All materials and sequences used can be found in Table 2 and Table 3.

From each qRT-PCR, melting curve, cycling threshold, and melting temperature were analyzed to verify specificity and quality of the reaction. In case of sample contamination or dysfunction of reaction components (enzymes, nucleotides, etc.) melting curves showed aberrant melting temperature or curve outline and qRT-PCR was redone. If more than four different miRNA sequences from one sample showed no valid data (no signal detectable or invalid melting curve), the whole saliva miRNA preparation was discarded. In this case, miRNA extraction from the remaining saliva sample was redone and qRT-PCR performed again. If no valid data could be acquired after three trials, the saliva sample was classified as contaminated and the patient was not considered for further miRNA expression analysis. Independently from that, if the U6 snRNA showed no detectable reaction or an invalid melting curve, the assay was discarded and redone as well. Valid qRT-PCR data sets provided cycle threshold (CT) values for further miRNA expression analysis.

In order to determine individual expression levels for each miRNA, we utilized the comparative ∆CT method [37]. qRT-PCR was performed on 96-well plates with three qRT-PCR mixes for each miRNA plus a triplet of the internal control (U6 snRNA). From these triplets mean values were calculated to eliminate statistical measurement errors. Potential interplate variations or differences are negligible in this method because the internal control gene serves as well as a standardizing primer for qRT-PCR quality.

In accordance with the comparative ∆CT method, mean CT values were transformed into ∆CT values depicting the individual expression level of each miRNA (∆CT = CT_internal control − CT_miRNA). If significant group differences were detected during statistical analysis, we described the expression level differences by foldchange determination as described by the comparative ∆CT method (foldchange = 2^ − (∆CT_internal control − ∆CT_miRNA). If ∆CT_internal control is greater than the ∆CT of a miRNA, the value of the equation is smaller than 1. In these cases, Schmittgen and Livak et al. [37] take the negative inverse of the ∆∆CT equation to provide the expression reduction as foldchange (foldchange = −1/(2^ − (∆CT_internal control − ∆CT_miRNA)).

### 2.3. Statistical Analysis

General clinical parameters (e.g., age, PSA serum concentration, prostate volume etc.) were checked for significant group differences between the cancer group and the no-cancer group. Due to sample insufficiencies (contamination, sample volume discrepancies, etc.), only 74 of the total 86 saliva samples were considered for further miRNA analysis. Statistical analysis focused on concentration differences of each miRNA between the cancer and no-cancer group and was performed using parametric or non-parametric tests depending on distribution character of the data. Unless stated otherwise, statistical analyses were conducted in SPSS version 25 (IBM, Armonk, NY, USA). *p*-values < 0.05 were considered significant. In case of significant group differences, the diagnostic accuracy of miRNA was verified using ROC curve analysis determining the area under the curve, cut-off value, sensitivity and specificity. Predictive values were calculated using absolute case numbers.

The cancer collective (*n* = 43) showed a broad range of Gleason Score classifications [see Table 4], but case numbers and distribution among subgroups was highly unbalanced and additional information concerning tumor localization, histological entity and metastasis status were partially incomplete. Therefore, we decided against further statistical analysis of differences between Gleason score subgroups concerning general clinical parameters as well as miRNA concentration.

**Table 4 biomolecules-12-01366-t004:** Statistical analysis of standardized screening parameter from prostate cancer screening program; descriptive analysis; MW: mean, SD: standard deviation, MDN: median; * indicates parameters with significant group differences (Prostate volume: U[31,43] = −3.213, *p* = 0.001, Mann–Whitney-U).

Participant Collective		Cancer Group	Control Group	Total
[N] (n/total)		43 (58%)	31 (42%)	74 (100%)
**Age (Years)**				
MW/SD		69.32/8.82	66.96/9.33	68.34/9.05
MDN		70.00	69.00	70.00
Range		41.00	39.00	42.00
**PSA (ng/mL)**				
MW/SD		30.84/70.49	11.20/17.43	22.61/55.49
MDN		8.63	6.80	8.40
Range		429.24	99.16	429.48
**fPSA (Unbound PSA) (ng/mL)**				
MW/SD		5.73/16.19	1.96/2.74	4.15/12.55
MDN		1.26	1.29	1.27
Range		98.75	15.32	98.75
**PSA Ratio (fPSA/PSA)**				
MW/SD		0.16/0.07	0.20/0.09	0.17/0.08
MDN		0.16	0.18	0.16
Range		0.29	0.38	0.42
**Prostate Volume (mL) ***				
MW/SD		43.09/22.90	69.45/40.57	54.13/33.91
MDN		37.00	52.00	45.00
Range		98.00	139.00	148.00
**Urine Culture with bacterial Growth [N] (n/Group)**				
Positive		10 (23%)	13 (42%)	25 (31%)
Negative		33 (77%)	18 (58%)	51 (69%)
**Gleason Score [N] (n/Cancer Group)**				
**6**	**7a**	**7b**	**8**	**9**
11 (26%)	14 (33%)	13 (30%)	1 (2%)	4 (9%)

## 3. Results

Established prostate cancer screening parameters like age or PSA serum concentration were documented for the examined participant collective and analyzed concerning significant differences between patients suffering from prostate cancer and the no-cancer control. No significant differences in age or PSA serum concentration were detected between both groups, except for prostate gland volume determined via transrectal ultrasound. Prostate glands were significantly smaller among patients of the cancer group (mean volume 43.09 mL, SD ± 22.90 mL) compared to the no-cancer control group (mean volume 69.45 mL, SD ± 40.57 mL) (U[31,43] = −3.213, *p* = 0.001, Mann–Whitney-U) [See Table 4].

All sixteen prostate cancer-associated miRNAs were detectable in exosomes from saliva samples processed via the modified sampling method by Wiegand et al. and were therefore part of further comparative analysis. hsa-mir-200b and hsa-mir-331-3p showed statistically significant concentration differences between cancer patients and the non-cancer control group (hsa-mir-200b: cancer group mean ∆CT = 3.27, SD ± 3.11 and non-cancer control group mean ∆CT = 5.11, SD ± 3.23, U[31,43] = −2.383, *p* = 0.017, Mann–Whitney-U; hsa-mir-331-3p: cancer group mean ∆CT = 1.78, SD ± 2.81 and non-cancer control group ∆CT = 3.17, SD ± 2.84, U[31,43] = −2.158, *p* = 0.031, Mann-Whitney-U) [38].

Figure 2 presents boxplots of expression levels of hsa-mir-200b and hsa-mir-331-3p between both groups. Hsa-mir-200b was 3.6-fold reduced and hsa-mir-331-3p was 2.64-fold reduced in cancer patients compared to patients suffering from other diseases of the urogenital tract.

Table 5 shows mean ∆CT values for each miRNA and fold changes between the cancer group and the non-cancer control group, respectively. The other 14 Prostate cancer associated miRNAs showed no significant difference in expression levels between both groups.

We performed ROC curve analysis to validate diagnostic accuracy for both significant miRNAs and determined an area under the curve of 0.663 for hsa-mir-200b and 0.648 for hsa-mir-331-3p [39], as Figure 3 and Figure 4 shows. Cut-off values were determined using Youden-Index and revealed best group differentiation at ∆CT = 5.5 for hsa-mir-200b with a sensitivity of 81% and specificity of 55% and at ∆CT = 2.8 for hsa-mir-331-3p with a sensitivity of 74% and specificity of 58% [40].

Positive and negative predictive values of both miRNAs were calculated using the valid cases of this study [see Table 6] [41]. hsa-mir-200b showed a positive predictive value of 71% to indicate prostate cancer if the ∆CT-value was greater than or equal to 5.5. miRNA hsa-mir-331-3p showed a positive predictive value of 71% to indicate prostate cancer when ∆CT-value was smaller than or equal to 2.87.

## 4. Discussion

In this study we aimed to find an easy-to-execute, non-invasive prostate cancer screening method based on salivary miRNAs to augment established screening tools such as PSA serum measurements, which among other insufficiencies produce high numbers of false positive prostate cancer suspects [42]. Therefore, we examined prostate cancer-associated miRNAs in easy-to-provide-and-collect saliva samples and hypothesized that concentrations of prostate cancer-associated miRNAs differ significantly between men suffering from prostate cancer and a non-cancer control group.

Two out of sixteen examined miRNAs (hsa-mir-331-3p, hsa-mir-200b) were significantly reduced in saliva samples from prostate cancer patients compared to the non-cancer control group. The observed group differences from saliva samples correspond with former studies reporting a significant reduction of hsa-mir-200b and hsa-mir-331-3p in prostate tumor cells compared to healthy prostate tissue samples [43,44,45,46].

Hsa-mir-200b is a member of the mir-200-family whose involvement in carcinogenesis and aberrant cell development is documented for many different cancer types. In prostate cancer, reduction or loss of hsa-mir-200b limits the suppression of Bmi-1, which is accountable for abnormal cell proliferation, migration and chemosensitivity [47]. Other research shows that downregulation of hsa-mir-200b can induce epithelial-mesenchymal transition (EMT) in prostate gland cells due to a loss of ZEB1 repression [48,49]. If ZEB1 is released, it inhibits E-cadherin expression. E-cadherin is one of the most important members of cell connection molecules such as desmosomes or tight junctions, and is therefore very important to EMT [50].

For hsa-mir-331-3p, former studies revealed an important tumor suppressor role in prostate cancer development. The findings of Epis et al. (2009) suggest a role for hsa-mir-331-3p in the development and progression of prostate cancer while focusing on ERBB-2 as a target of this miRNA [45]. A loss of hsa-mir-331-3p expression could promote the increased ERBB-2 expression and signaling seen in many prostate cancers [45]. Wang et al. (2009) found hsa-mir-331-3p to be differentially expressed in prostate cancer cell lines and implicated an important part that hsa-mir-331-3p plays in cell cycle regulation [46]. Decreased levels of hsa-mir-331-3p concur with former studies that mostly showed decreased levels of hsa-mir-331-3p in prostate cancer patients [51]. Former studies showing results that indicate a downregulation of hsa-mir-331-3p were studying cell lines rather than body fluids. Shee et al. provided the most recent review of the pathological and physiological roles of the miRNA-331 family in cancer [52].

As described under the methods section, saliva processing and analysis were based on the protocol presented by Gallo et al. with modifications established by Wiegand et al. In the original paper of Gallo et al. saliva samples were immediately centrifuged and purified to prevent secondary lysis of cells and/or other cellular residues (e.g., mucosa cells, bacteria, food debris). However, these first steps especially are the most difficult to execute in smaller healthcare institutions, practices or for non-professional individuals at home. The modified sampling and storage protocol established by Wiegand et al. [32] freezes the saliva specimens at −80 °C after collection before the first centrifugation and purification steps of Gallo et al. Exosome stability beyond a freeze thawing process has already been proven by Gallo et al., but an addition of the protocol by freeze storing was feared to subsequently release different microvesicles into the saliva complex and potentially interfere with the following exosomal miRNA analysis [16].

Wiegand’s freeze storing modification could enhance the possibility of a widespread use of salivary miRNA analysis in everyday clinical life and even the outpatient sector, since this modified sampling process is much easier to implement. Saliva is naturally contaminated with the remains of cell debris, food and lysed bacteria, which probably account for an unspecific fraction of microvesicles in the body fluid. Freeze thawing of the samples will undeniably add different kinds of intracellular microvesicles to the primarily desired fraction of extracellular salivary exosomes. We hypothesized that miRNA analysis focusing on prostate cancer associated miRNAs would withstand any major interferences such as sample contamination or secondary cell lysis by virtue of the sequence’s disease specific expression. Gallo et al. themselves stated that even though the analysis of only extracellular exosome-derived miRNA enhances the sensitivity of miRNA, it is not mandatory [16].

This study supports their statement and showed that concentration measurements of prostate cancer-associated miRNAs (i.e., in this study hsa-mir-200b and hsa-mir-331-3p) extracted from salivary exosomes (regardless intracellular or extracellular exosome) show significant group differences between prostate cancer participants and the non-cancer control group with reliable diagnostic value. Thus, the effects of freeze thawing seem to be minor and negligible in case of disease-specific miRNA analysis. The reliability of the results is underpinned by the prostate cancer associated expression of those two miRNA sequences numerously described in other studies [9,10,11,43,44,45].

Further investigation of the diagnostic strength of hsa-mir-200b and hsa-mir-331-3p was determined via ROC curve analysis, which additionally allows a determination of a cut off value with an ideal ratio of sensitivity to specificity. Both miRNAs presented moderate but reliable strength in differentiating between cancer and the non-cancer control group (AUC of 0.663 for hsa-mir-200b and 0.648 for hsa-mir-331-3p) [39] [see Figure 2]. This is consistent with Souza et al. (2017), who described similar diagnostic values for hsa-mir-200b (AUC of 0.57, sensitivity = 67%, specificity = 75%) although examining concentration differences in blood serum samples rather than saliva exosomes [53]. Application of the determined cut off values revealed that miRNA concentration in saliva exosomes was consistent with the histologic findings from prostate biopsies in about 70% of the cases, indicating a positive predictive value of 71% for both sequences to indicate prostate cancer [see Table 4]. Classification of the miRNA test results according to the cut-off value (i.e., expression level indicates prostate cancer or no-cancer) revealed that in most cases, miRNA expression in both sequences was altered in the same direction—either 200b+/331+ or 200b−/331−. Only 15 patients showed contrary test results, meaning that one miRNA indicated prostate cancer while the other miRNA indicated no cancer. The cumulation of double positive test results may hint at the existence of a diagnostic pattern of different miRNA sequences, possibly with higher diagnostic value for prostate cancer than the diagnostic ability of one miRNA sequence alone. To investigate this hypothesis, we compared sensitivity and specificity of a double positive or double negative test result to the individual diagnostic properties of hsa-mir-200b or hsa-mir331. However, we found no significantly higher values of a combined miRNA examination over a single sequence approach. Even if miRNA patterns of higher diagnostic accuracy existed, reduction of analytical and methodological complexity as well as susceptibility to errors would probably favor a single-sequence approach in any possible future application. For a profound and certain answer to this question, further research and comparison of prostate cancer-associated miRNA in saliva exosomes is needed to determine a sequence with the highest possible diagnostic ability.

For comparison, meta-analysis of PSA serum tests (cut off value 4 ng/dL) determined a positive predictive value for prostate cancer of only 30% (negative predictive value: 85%) [3]. Although the comparison of diagnostic values indicates a superior diagnostic ability of miRNA concentration measurements over established PSA serum measurements, a thorough comparison of both methods is inadmissible at the moment. The diagnostic strength of PSA serum tests has been studied using a broad participant collective from the general population with the aim to identify prostate cancer patients among healthy men. The presented study recruited a pre-screened participant collective of men with elevated PSA levels and investigated miRNA expression differences to identify prostate cancer patients among men with abnormal PSA serum test results. Nevertheless, hsa-mir-200b and hsa-mir-331-3p extracted from salivary exosomes demonstrated a reliable diagnostic strength to differentiate between cancer and non-cancer patients. A differentiation based on PSA values of the same participant collective proved insufficient in this study.

Fourteen other miRNA sequences were examined and analyzed in this study. All sequences have successfully been detected in salivary exosomes but in contrast to has-mir-200b and has-mir-331-3p without significant concentration differences between cancer and non-cancer participants. The reason for this inconsistency remains unclear as they all have been described as prostate cancer-associated sequences in the preliminary literature research [19,29,30,31]. Each miRNA was determined by consulting literature examining prostate cancer-associated miRNAs. Those studies examined different materials such as prostate tissue, prostate cancer cell lines or blood serum samples rather than salivary exosomes. Detailed studies on miRNA in saliva are limited and many factors influencing miRNA concentration such as saliva filtration processes or active miRNA equipment of salivary exosomes are unknown and could provide information on further regulation of these 14 miRNAs in saliva or its exosomes [16,54,55].

Another reason may be found in the composition of the no-cancer control group of this study. All participants recruited for this study were men suspected to suffer from prostate cancer due to elevated PSA serum levels and admitted to hospital for prostate biopsy to histologically secure or disprove a cancer diagnosis. Men diagnosed with benign diseases of the urogenital tract (such as prostatitis, urinary occlusion, urinary tract infection but most commonly benign prostatic syndrome) later formed the non-cancer control group. Dysregulated miRNA leading to abnormal cell-cycle, cell division or cell metabolism may eventually cause tissue carcinogenesis but may similarly be involved in benign prostate pathologies such as prostatitis or benign prostate hyperplasia, e.g., miRNA-106a in BPH [56]. Thus, the 14 other sequences might indicate urogenital pathologies in general rather than solely prostate cancer and therefore show no significant group differences in this study. Future research needs to identify prostate cancer-associated miRNA and verify their specificity in comparison to other diseases of the urogenital tract.

The present study produced promising results for a saliva-based miRNA test method in prostate cancer diagnostics. The study design included men with elevated PSA serum levels. Half of the patients were diagnosed with prostate cancer; the others were diagnosed with benign diseases of the urogenital tract accompanied by PSA level elevation. The results demonstrated a reliable identification of cancer patients among prostate cancer suspects. This hints at a high potential to identify prostate cancer patients in general prostate cancer screenings. However, healthy individuals were not included in the study and are not part of the comparative analysis which is why a detailed diagnostic value has yet to be examined. Nevertheless, the present study design highlights a problematic aspect of the current prostate cancer screening program and presents salivary miRNA measurement as a possible solution. The current prostate cancer screening produces high numbers of false positive cancer suspects, mainly due to the low predictive value of PSA serum measurements [57]. Other current diagnostic tools such as the mpMRI, SelectMDx test, and of course, urinary biomarkers are definitely contributing to a more diverse range of choices [58]. Keeping in mind that prostate cancer is a global phenomenon, diagnostic tools should be easily accessible. Urinary miRNAs are a serious alternative, but could possibly be difficult to obtain if there is pre-existing urinary tract disease [11]. Furthermore, it is not claimed that salivary miRNAs are the only option, but rather a complementary factor.

This study showed a reliable differentiation of prostate cancer patients and those falsely suspected to suffer from prostate cancer due to elevated PSA levels (no-cancer control group) via non-invasive measurement of hsa-mir-331-3p and/or hsa-mir-200b in exosomes from whole saliva samples. Due to the differentiation ability of those two miRNA sequences, a conspicuous PSA test result could be augmented by this method to verify or refute an uncertain cancer suspicion. Thanks to the modified sampling and processing protocol, this augmentative diagnostic method could be applied in nearly every healthcare institution, in the outpatient sector and even at home. In consequence, hsa-mir-331-3p and hsa-mir-200b concentration measurement in saliva samples could reduce numbers of false positive cancer suspects in the preventive screening program and thereby reduce numbers of falsely indicated prostate biopsies.

## 5. Conclusions

All recruited patients of the present study were suspected to suffer from prostate cancer due to elevated PSA serum levels in preventive screening procedures. In the end, only half of these suspects were diagnosed with malignant prostate cancer; the others showed elevated PSA levels due to benign reasons of other kinds. A differentiation of both groups based on the standard screening parameters and their assessment by medical experts proved insufficient. The present study demonstrated for the first time an easy-to-execute and non-invasive screening method based on salivary miRNA with the ability to substantiate or contradict prostate cancer suspicion with a prognostic value of 70%.

Saliva samples could be easily collected and frozen in any healthcare institution and further processed in centralized laboratories. An easily accessible tool could advise clinicians and/or patients in the decision-making process for further diagnostic steps, potentially reduce the number of false positive prostate biopsies, and in the end, may increase the overall sensitivity and specificity of prostate cancer screenings.

These findings suggest that exosome-derived salivary miRNA could serve as a reliable and, most importantly, non-invasive evaluation tool of prostate cancer screening results (e.g., PSA serum measurements) under the concept of liquid biopsy [59].

We believe that further investigation of salivary exosomes and miRNA expression assays might reveal more potential candidates of prostate cancer-associated circulating miRNAs enabling a non-invasive, saliva-based test method with high prognostic value to indicate prostate cancer and maybe replace PSA measurements in the long run. In this context, hsa-mir-200b and hsa-mir-331-3p are the first two promising candidates for an additional non-invasive part in prostate cancer diagnostics.

## Figures and Tables

**Figure 1 biomolecules-12-01366-f001:**
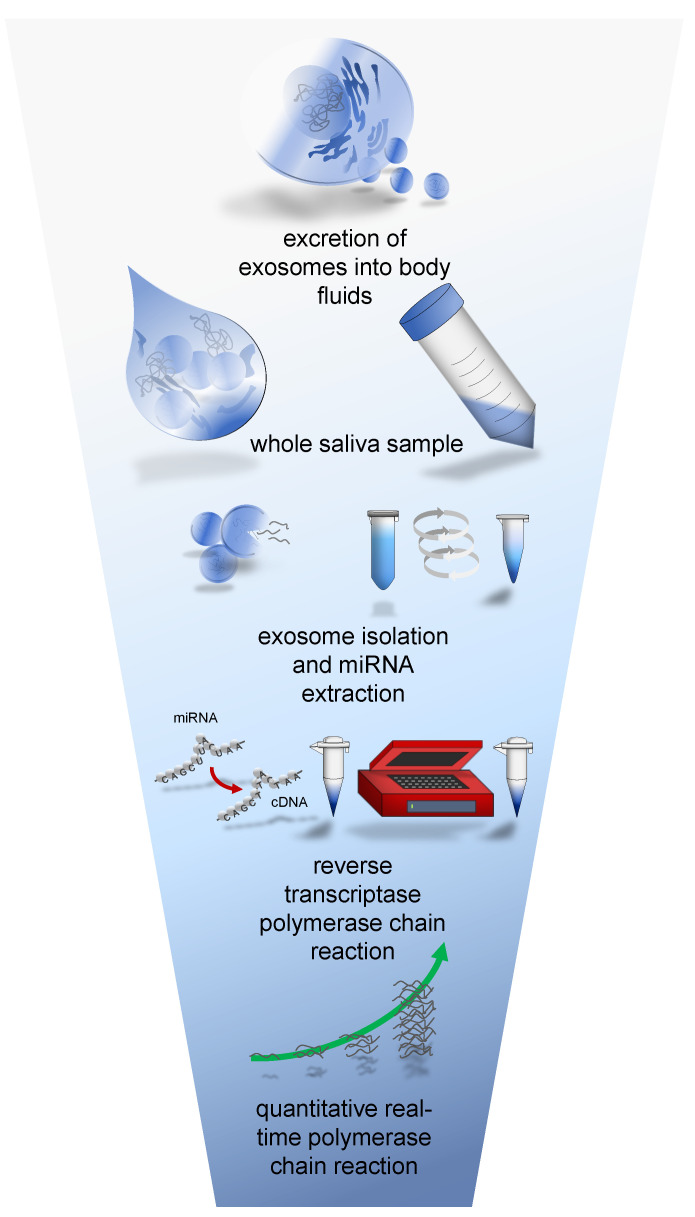
Overview of salivary miRNA analysis with simplified preparation steps from whole saliva sampling till qRT-PCR analysis.

**Figure 2 biomolecules-12-01366-f002:**
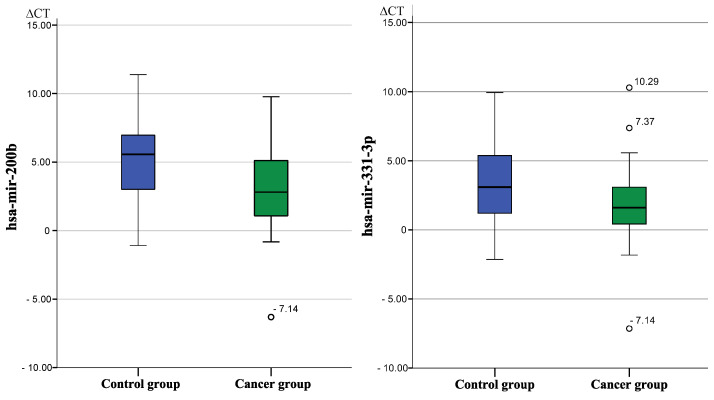
Boxplots of hsa-mir-200b and hsa-mir-331-3p with significant expression level differences between the non-cancer control group and the cancer group.

**Figure 3 biomolecules-12-01366-f003:**
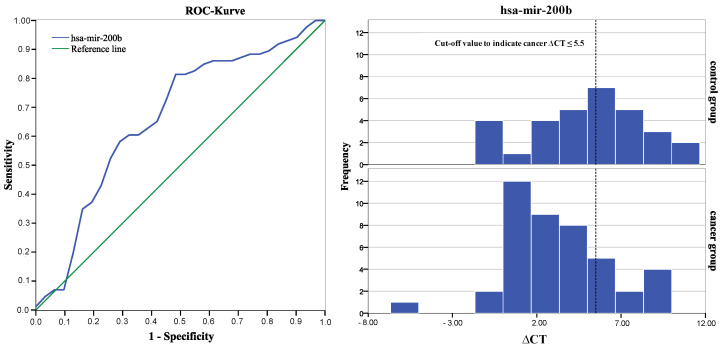
ROC curve of miRNA miR-200b with an area under the curve (AUC) of 0.663 (standard error 0.066, 95% confidence interval 0.53–0.79, asymptotic significance α = 0.014). The histogram compares numbers of participants with equal ∆CT values from each group. The dotted line depicts the determined cut-off value and separates true positives (cancer group = 35 of 43) and false positives (no-cancer group = 14 of 31) from true negatives and false negatives.

**Figure 4 biomolecules-12-01366-f004:**
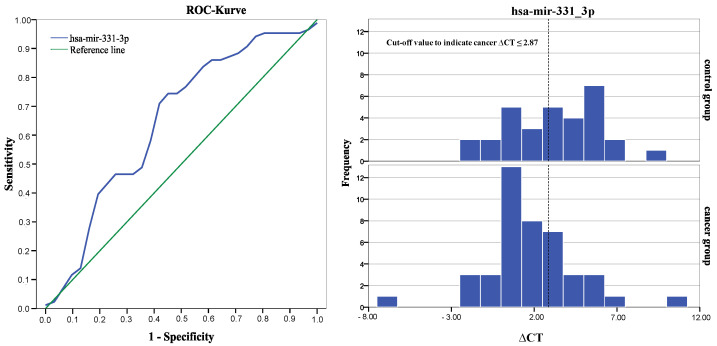
ROC curve of miRNA miR-331-3p with an area under the curve (AUC) of 0.648 (standard error 0.067, 95% confidence interval 0.52–0.78, asymptotic significance α = 0.028). The histogram compares numbers of participants with equal ∆CT values from each group. The dotted line depicts the determined cut-off value and separates true positives (cancer group = 32 of 43) and false positives (non-cancer group = 13 of 31) from true negatives and false negatives.

**Table 1 biomolecules-12-01366-t001:** Sources of literature and miRNAs withdrawn from these sources.

Source of Literature	miRNA
Chen et al. (2012) [19]	hsa-mir: 622
Bryant et al. (2012) [29]	hsa-mir: 574-3p, 625, 331-3p, 141, 130b, 432, 484, 375, 107, 181a, 2110, 301a, 326
Brase et al. (2011) [30]	hsa-mir: 200b, 141, 375
Moltzahn et al. (2011) [31]	hsa-mir: 106a

**Table 2 biomolecules-12-01366-t002:** Vendors of equipment, chemicals and software used for this study.

Equipment	
**PCR tower**	Jena Bioscience, Jena, Germany
**Sorvall MGX-120 Ultracentrifuge**	Thermo Fisher Scientific, Waltham, MA, USA

**Chemicals**	
**TRIzol™ Reagent**	Thermo Fisher Scientific, Waltham, MA, USA
**Chloroform**	Merck, Darmstadt, Germany
**Isopropanol C_3_H_8_O**	Merck, Darmstadt, Germany
**RNAse free H_2_O**	Thermo Fisher Scientific, Waltham, MA, USA
**Ethanol (75%)**	Merck, Darmstadt, Germany

**Software**	
**qPCR Soft**	Thermo Fisher Scientific, Waltham, MA, USA
**Microsoft Office**	Microsoft, Redmond, WA, USA
**SPSS**	IBM, Armonk, NY, USA
**Endnote**	Thomson Reuters, Toronto, ON, Canada

**Table 3 biomolecules-12-01366-t003:** Primer assays ID and sequence, acquired from Qiagen^®^.

microRNA	Mature	Accession	Sequence
**MIR106A**	hsa-miR-106a-5p	MIMAT0000103	13-AAAAGUGCUUACAGUGCAGGUAG-35
**MIR130B**	hsa-miR-130b-5p	MIMAT0004680	13-ACUCUUUCCCUGUUGCACUAC-33
**MIR301A**	hsa-miR-301a-5p	MIMAT0022696	14-GCUCUGACUUUAUUGCACUACU-35
**MIR331**	hsa-miR-331	MIMAT0000760	61-GCCCCUGGGCCUAUCCUAGAA-81
**MIR326**	hsa-miR-326	MIMAT0000756	60-CCUCUGGGCCCUUCCUCCAG-79
**MIR375**	hsa-miR-375-3p	MIMAT0000728	40-UUUGUUCGUUCGGCUCGCGUGA-61
**MIR484**	hsa-miR-484	MIMAT0002174	8-UCAGGCUCAGUCCCCUCCCGAU-29
**MIR2110**	hsa-miR-2110	MIMAT0010133	8-UUGGGGAAACGGCCGCUGAGUG-29

**Table 5 biomolecules-12-01366-t005:** Summarized mean ∆CT values for each miRNA within the cancer group and the non-cancer control group and foldchanges, respectively. *p*-values of ≤ 0.05 were considered as significant results. Mann–Whitney-U; SD: standard deviation; * indicates parameters with significant group differences.

PCa-Specific microRNA	∆CT Cancer Group (Mean/SD)	∆CT Control Group (Mean/SD)	Foldchange (Control Group—Cancer Group)	*p*-Value
**hsa-mir-200b ***	**3.27/3.11**	**5.12/3.23**	**−** **3.60**	**0.017**
**hsa-mir-331-3p ***	**1.78/2.81**	**3.17/2.84**	**−** **2.64**	**0.031**
hsa-mir-107	1.41/3.38	2.34/3.84	No significant difference	0.290
hsa-mir-141	0.83/2.66	1.47/2.95	No significant difference	0.224
hsa-mir-432	2.87/2.71	3.46/3.05	No significant difference	0.446
hsa-mir-574	5.83/3.40	6.00/2.36	No significant difference	0.874
hsa-mir-625	1.30/3.00	2.41/3.45	No significant difference	0.174
hsa-mir-181	1.78/3.81	1.99/3.19	No significant difference	0.657
hsa-mir-622	1.79/2.81	3.17/2.84	No significant difference	0.890
hsa-mir-375	1.56/4.47	0.97/4.40	No significant difference	0.806
hsa-mir-484	3.22/3.90	2.80/3.10	No significant difference	0.766
hsa-mir-2110	1.55/4.49	2.47/2.70	No significant difference	0.433
hsa-mir-130b	1.39/3.66	1.82/2.87	No significant difference	0.552
hsa-mir-301a	3.27/3.55	2.98/2.67	No significant difference	0.739
hsa-mir-326	6.12/3.29	5.87/2.47	No significant difference	0.959
hsa-mir-106a	5.00/3.33	5.19/2.77	No significant difference	0.782

**Table 6 biomolecules-12-01366-t006:** Calculation of predictive values for miRNA hsa-mir-200b and hsa-mir-331-3p.

hsa-mir-200b, Cut off ∆CT = 5.5	Tested Positive (X ≤ 5.5)	Tested Negative (X ≥ 5.5)	
cancer group [N] = 43	35	8	sensitivity0.814
control group [N] = 31	14	17	specificity0.548
Predictive value	0.714	0.680	

**hsa-mir-331,** **cut off ∆CT = 2.87**	tested positive (X ≤ 2.87)	tested negative (X ≥ 2.87)	
cancer group [N] = 43	32	11	sensitivity0.744
control group [N] = 31	13	18	specificity0.581
predictive value	0.711	0.462	

## Data Availability

The datasets used and/or analyzed during the current study are available from the corresponding author on reasonable request.

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
