# Peer review of "Prostate Cancer-Associated miRNAs in Saliva: First Steps to an Easily Accessible and Reliable Screening Tool"

_biomolecules, 2022, doi:10.3390/biom12101366_

Round 1

Reviewer 1 Report

Study presented by Luedemann and co-authors is very interesting where they used non-invasive method for screening of prostate cancer through microRNAs. Although this paper is having potential to contribute in the field of cancer research but there are some serious issues that can not be neglected and require major revision of the paper.

Flow of the paper should be

Introduction

Materials and method

Results

Discussion

Conclusion

References

Flow of paper is not up to the mark and require re-write of various sections of abstract, introduction, results and discussion.

Presentation of the paper is very ambiguous and lack of uniformity throughout the paper.

Need to re-work on diagrams and tables, make them simple.

References require special care and re-write all references according to the journal’s guidelines.

There are lots of issue on the grammar and sentence formation, correction is warranted.

Author Response

Dear reviewer 1,

thank you very much for your valuable comments. Please see the attachment.

Thank you

Reviewer 2 Report

Thanks for your contribution.

Please find my comments in the attached file 

Author Response

Dear reviewer 2,

thank you very much for your valuable comments. Please see the attachment.

Thank you

Reviewer 3 Report

In this study, men suspected of harboring prostate cancer (due to raised serum PSA levels) were assessed for selected microRNAs levels in saliva samples, to determine whether these might serve as biomarkers for early detection of prostate cancer. Of 16 candidate miRs, searched in the literature published until 2014, only 2 disclosed diagnostic value. Overall, in a series of 74 cases, of which 43 correspond to men with prostate cancer diagnosed in prostate biopsy and the remainder 31 corresponding to men in which no prostate cancer was found, the sensitivity of salivary (exosomal) hsa-mir-200b and hsa-mir-331-3p was about 75-80% and specificity inferior to 60%.

The main strength of the manuscript is the novelty of using salivary miRs for prostate cancer detection. Nonetheless, the study has several important flaws that undermine the conclusions. Specifically:

1. The authors state that the literature search for "prostate cancer-specific" miRs included papers published until January 2014. Considering that we are in 2022, this does not seem reasonable nor justifiable. In the same vein, looking at the References, it becomes clear that several are outdated.

2. The two miRs advocated as "prostate cancer-specific" - hsa-mir-200b and hsa-mir-331-3p - are not specific ta all. For instance, hsa-mir-200b is involved in EMT and , thus, it is very commonly deregulated in several cancer types (e.g., renal, breast, bladder, prostate), including some that may directly contribute with exosomes to saliva (esophageal, oral cavity, ...).

3. Considering that patients were enrolled based on altered serum PSA levels, the results do not provide evidence that this test might be used for early detection of prostate cancer, eventually replacing serum PSA, but rather as a complement of PSA testing.

4. Considering the Materials and Methods, it seems that no normalization for input was performed and that "raw" expression values were used for comparison among the groups. Considering that qRT-PCR was used to assess candidate miRs expression, normalization is mandatory.

5. No mention is made as to whether informed consent was obtained from the study participants and if the project was assessed and approved by an Ethics Committee.

6. The Abstract is too long and the Introduction is too small, not providing an adequate contextualization of the study.

7. It is not clear why the authors did not tested the use of both markers hsa-mir-200b and hsa-mir-331-3p simultaneously to verify whether it migh improve the performance.

Author Response

Dear reviewer 3,

thank you very much for your valuable comments. Please see the attachment.

Thank you

Round 2

Reviewer 1 Report

Corrections has been incorporated in revised manuscript. 

Reviewer 2 Report

Dear authors,

Thanks for considering to my raised points from the first submission.

Nevertheless, there is still a point which need to be solved and improve before acceptance for publication.

Moving the two first paragraph from Materials and Methos to the introduction can’t be done just as it is. It must be done by rewriting the introduction part to make it smooth and easy to read. It is more a question about the idea or concept which must be part of the introduction than moving lines without sense.

Hoping you can address these points.

All the best,

Reviewer 3 Report

The authors have performed signficant improvements in the manuscript, which should be acknowledged.

Nonetheless, concerns remain and need clarification:

1) In the methods description of the performed qPCR approach there are some important information still missing (e.g., primers assays ID and vendor used to test the miRNAs expression as well as which equipment was used to generate the results). The authors should provide a table with all this data, which is fundamental to allow replication of the results by other labs.

2) In the Results section, the graphs representing miRNAs expression in the different groups describe only the delta Ct value. Nevertheless, these values are a bit confusing and do not show the reduction calculated by fold-change. Actually, at first sight and without paying attention to the graph's description, it seems that the expression is higher in cancer compared to control group, whereas, in fact it, is the opposite. The authors should consider alter the way how they present the results in the graphs.

3) Likewise, in material and methods, the authors mention that qPCR analysis was done by ddCt, but only results of dCt are presented in the manuscript. This raises some concerns on whether the authors had or not in consideration the interplate variation and if so, how was it performed.
